# Fetal Growth and Neonatal Outcomes in Pregestational Diabetes Mellitus in a Population with a High Prevalence of Diabetes

**DOI:** 10.3390/jpm12081320

**Published:** 2022-08-16

**Authors:** Giampiero Capobianco, Alessandra Gulotta, Giulio Tupponi, Francesco Dessole, Giuseppe Virdis, Claudio Cherchi, Davide De Vita, Marco Petrillo, Giorgio Olzai, Roberto Antonucci, Laura Saderi, Pier Luigi Cherchi, Salvatore Dessole, Giovanni Sotgiu

**Affiliations:** 1Gynecologic and Obstetric Clinic, Department of Medicine, Surgery and Pharmacy, University of Sassari, 07100 Sassari, Italy; 2Pediatric Pulmonology and Respiratory Intermediate Care Unit, Sleep and Long-Term Ventilation Unit, Department of Pediatrics, Bambino Gesù Children’s Hospital, IRCCS, 00143 Rome, Italy; 3Chronic Pelvic Pain Centre, Dep. of Obstetrics and Gynaecology, Ospedale S. Maria Della Speranza, 84091 Battipaglia, Italy; 4Neonatal Intensive Care Unit (NICU), Sassari University, 07100 Sassari, Italy; 5Pediatric Clinic, Sassari University, 07100 Sassari, Italy; 6Clinical Epidemiology and Medical Statistics Unit, Department of Medical, Surgical and Experimental Sciences, University of Sassari, 07100 Sassari, Italy

**Keywords:** pregestational diabetes mellitus (PGDM), type 1 diabetes mellitus (DM1), type 2 diabetes mellitus (DM2), maturity onset diabetes of the young (MODY), fetal outcomes, neonatal outcomes

## Abstract

The aim of this retrospective study, conducted in an Italian tertiary care hospital, was to evaluate maternal-fetal and neonatal clinical outcomes in a group of patients with pregestational diabetes mellitus (PGDM), such as diabetes mellitus type 1 (DM1), diabetes mellitus type 2 (DM2), and maturity onset diabetes of the young (MODY). Overall, 174 pregnant women, nulliparous and multiparous, with a single pregnancy were enrolled. Data on pregnancy, childbirth, and newborns were collected from medical records. The selected patients were divided into two groups: the PGDM group (42 with DM1, 14 with DM2, and 2 with MODY), and the control group (116 patients with a negative pathological history of diabetes mellitus). We reported an incidence of preterm delivery of 55.2% in the PGDM group, including 59.5% of those with DM1 and 42.9% of those with DM2, vs. 6% in the controls. Fetal growth disorders, such as intrauterine growth retardation, small for gestational age, and fetal macrosomia were found in 19% and 3.6% in the case and control groups, respectively. A relationship between DM2 and gestational hypertension was found.

## 1. Introduction

The prevalence of diabetes mellitus in pregnancy is increasing worldwide in parallel with that of obesity. Gestational diabetes mellitus (GDM) is diagnosed in the majority of the cases, followed by pre-gestational diabetes (PGDM), such as diabetes mellitus type 1 (DM1) and type 2 (DM2) [1].

The Italian region Sardinia has an incidence rate of DM1 equal to 33.4 per 100,000, which is the second highest globally [2]. Pre-gestational diabetes is associated with adverse neonatal outcomes [3].

The incidence of adverse maternal outcomes is high in case of PGDM [4,5,6,7,8]: abortions and low birth weight (<2500 g) were more common, as well as congenital anomalies. Notably, the types and patterns of congenital malformations associated with maternal PGDM are non-random [9], with an increased risk of heart, central nervous system, and skeleton malformations.

The data in the literature state that impaired glucose metabolism, such as that experienced by patients with PDGM, may represent a risk factor for the onset of numerous pregnancy complications and may be associated with adverse neonatal outcomes. The aim of the study was to fully understand the complications of pregnancy in patients with PDGM, to prevent such complications, to ensure better obstetric care during pregnancy and childbirth, and to improve neonatal outcomes in a region with a high prevalence of this metabolic pathology.

Impaired glucose metabolism could be a risk factor for pregnancy complications and may be associated with adverse neonatal outcomes [1,2,3,4]; this study evaluated the maternal–fetal and neonatal clinical outcomes of a cohort of patients with PGDM (DM1, DM2, and maturity onset diabetes of the young—MODY) in comparison with those of pregnant individuals without diabetes.

The aim of the study was to describe pregnancy complications in patients with PDGM living in a high prevalence Italian region.

## 2. Methods

A retrospective study was carried out: patients aged 18 and 44 years were enrolled between January 2016 and August 2020. They were followed-up in a tertiary care Italian hospital. A formal ethical approval was not needed according to the Italian law on observational studies.

Selected patients were divided into two groups: 58 patients formed the PGDM case group, of which 42 were diagnosed with diabetes mellitus type 1 (DM1), 14 with diabetes mellitus type 2 (DM2), and 2 with maturity onset diabetes of the young (MODY); 116 patients formed the control group, all of whom had a negative pathological history of diabetes mellitus and a negative 75 g oral glucose test tolerance (75 g OGTT) performed at 24–28 weeks of gestation [10,11]. The control group was randomly selected to avoid selection bias.

The two groups were matched by age (calculated at the time of delivery), with a ratio of 1:2 (PDGM: controls). The criteria for the diagnosis of diabetes included fasting plasma glucose (FPG) levels ≥ 126 mg/dL (7.0 mmol/L) and 2 h plasma glucose (PG) level ≥ 200 mg/dL (11.1 mmol/L) during a 75 g OGTT. Inclusion criteria were: pregnant women with pre-gestational diabetes (PGDM), such as diabetes mellitus type 1 (DM1) and type 2 (DM2). The exclusion criteria were diagnosis of gestational diabetes, twin pregnancy, and stillbirth.

Characteristics of pregnancy and delivery were collected at the hospital admission and at delivery. Neonatal data were retrieved from the admission registries and from the medical records of newborns admitted to the neonatal intensive care unit (NICU). The following data were collected: age, parity, height, pregravidic weight, weight at delivery, last menstruation, comorbidity (cardiovascular diseases, thyroid diseases, multiple sclerosis, and other autoimmune diseases), and prenatal screening surveys (combined test, noninvasive prenatal test, villocentesis, amniocentesis, and fetal echocardiography).

The variables collected for each group of women were summarized in the following categories:Pregnancy outcomes—gestational age at childbirth (GA); hospital stay; mode of delivery (spontaneous vaginal delivery or caesarean section).Diseases of pregnancy and fetal pathologies—threatened abortion (vaginal bleeding and symptoms that suggest that a woman is at an increased risk of miscarriage. Threatened preterm labor is the progression of cervical dilatation and ripening caused by regular uterine contractions occurring before 37 weeks of pregnancy, which may result in preterm birth); gestational hypertension; preeclampsia and HELLP syndrome; placental abruption; pathology of amniotic fluid (oligohydramnios and polydramnios); premature rupture of membranes (PROM) and preterm rupture of membranes (P-PROM); macrosomia; fetal growth restriction (FGR); morphological abnormalities diagnosed on ultrasound.Neonatal outcomes—weight at birth compared to those expected for the gestational age (in percentiles), and then classification within one of the classes of appropriate for gestational age (AGA), small for gestational age (SGA), or large for gestational age (LGA). For this study we used the definition of the Royal College of Obstetricians and Gynaecologists (RCOG) [12] which informs UK clinical practice, based on sonographic estimated fetal weight (EFW) measurement < 10th percentile to describe a fetus that has not reached its target weight. Patients were divided in three groups for comparison; fetuses with EFW < 10th percentile for gestational age (SGA), fetuses with EFW > 10th percentile for gestation (AGA) and fetuses > 90th percentile for gestation (LGA) according to the Alexander growth standard [13]; Apgar at the first minute and fifth minute; number of hospitalization days and at which intensity of care (nursery, neonatology or NICU); recognition of respiratory diseases at birth such as respiratory distress syndrome (RDS), transient tachycardia of the newborn (TTN) or apnea crisis, and if there has been any intubation; blood glucose at the third hour; hypoglycemia status and glucose supplementation; neonatal jaundice, treated or not with phototherapy; morphological abnormalities found at birth.

An ad hoc electronic database was created to collect all study variables. Qualitative data are summarized with absolute and relative (percentage) frequencies. Medians and interquartile ranges were used for quantitative variables with a non-parametric distribution. Chi-squared or Fisher’s exact test were used to compare qualitative variables for individuals with and without diabetes, whereas the Mann–Whitney test was used to compare non-normal quantitative variables. A two-tailed *p*-value < 0.05 was considered statistically significant. Statistical software STATA version 16 (StataCorp, College Station, TX, USA) was adopted for all statistical analyses.

## 3. Results

A total of 58 PGDM patients were recruited, of which 42 were diagnosed with DM1, 14 with DM2, and 2 with MODY. A further 116 patients with a negative pathological history of diabetes mellitus were recruited for the control group.

A higher median (IQR) pre-gestational body weight (61 (55.5–72.5) vs. 57 (50.5–63); *p*-value: 0.003] and BMI [23.7 (20.8–28) vs. 22 (19.8–24); *p*-value: 0.005] were found in the PGDM group (Table 1). Furthermore, the prevalence of obesity was significantly higher in the PGDM group than the control group.

A history of multiple sclerosis (5.2%) and Hashimoto’s thyroiditis (17.2%) was discovered in patients belonging to the PGDM group.

Among the patients in the PGDM group, 77.6% had a normal glycemia at the time of delivery, according to the values established by the American Diabetes Association (fasting glucose lower than 90 mg/dL and Hb glycated lower than 6%) [11].

### 3.1. Pregnancy Disorders

No statistically significant differences were found for the following outcomes: threatened abortion, abnormal placental insertion and detachment, amniotic fluid disorders, however, an association between the threatened abortion and DM2 was found. The frequency of threatened preterm labor in the PGDM group (24.1%) was higher than that in the control group (9.5%; *p*-value: 0.009). This difference was more striking when the incidence of preterm delivery was evaluated in DM1 patients (26.2%, *p*-value: 0.02). Pregnancy-induced hypertension and preeclampsia were reported only in the PGDM group. Amniotic fluid disorders were only detected in patients with DM1 (12.1%).

### 3.2. Fetal Disorders

Fetal growth disorders were more prevalent in the PGDM group (19%; *p*-value < 0.0001). Fetal macrosomia (fetal growth ≥ 95th percentile) was found in fetuses of diabetic mothers (*p*-value < 0.0001), the majority of whom were diagnosed with DM1. Fetal growth restriction (FGR) (fetal growth < 5th percentile) was found more frequently in patients with DM2 (21.4%; *p*-value: 0.02). Fetal echocardiography was requested more frequently in the PGDM group than in the control group (67.4% vs. 17%; *p*-value < 0.0001) to detect fetal cardiac abnormalities.

### 3.3. Pregnancy Outcomes

CS delivery was more frequently performed in PGDM patients (87.9% vs. 44.8%; *p*-value < 0.0001). This finding was confirmed by stratifying PGDM patients by DM (DM1: 90.5%; DM2: 85.7%). CS in emergency was more prevalent in the PGDM group (62.8% vs. 38.5%; *p*-value: 0.02). The PGDM group did not show a higher frequency of medical induction of vaginal delivery than that of the control group. Frequency of preterm deliveries was higher in the PGDM group (55.2% vs. 6.0%; *p*-value < 0.0001).

### 3.4. Neonatal Outcomes

The median length of hospital stay was three days for the births in the control group, eleven days for patients with DM1 (*p*-value < 0.0001), and six for those with DM2 (*p*-value: 0.0001) (Table 2).

Admission to NICU occurred more frequently in the PGDM group (*p*-value < 0.0001), with 73.8% of the births of the DM1 patients and 50% of the DM2 patients (*p*-value < 0.0001). Prevalence of respiratory disorders was higher the PGDM group (31%; *p*-value < 0.0001). Similarly, RDS and neonatal hypoglycemia occurred more frequently in the PGDM group. Incidence of neonatal jaundice was significantly higher (74.1%) in the PGDM cohort.

There was a statistically significant difference in birth weight centile values classified into fetus AGA (appropriate for gestational age), SGA (small for gestational age) and LGA (large for gestational age). The proportion of AGA was significantly higher in the control group (82.8% vs. 46.6%; *p*-value < 0.0001), whereas LGA prevalence was significantly higher in the PGDM group (46.5% vs. 6.9%; *p*-value < 0.0001), especially for patients with DM1 (57.1%; *p*-value < 0.0001).

Morphological anomalies were detected in 13.8% and 32.8% in the control and PGDM group (*p*-value: 0.003), respectively (Table 3).

## 4. Discussion

Diabetic patients had a pregravidic body weight higher than that of the patients in the control group, and the median pregravidic BMI differed by 1.7 points between cases and controls. In agreement with other studies [3,4,5,6,7,8], the increase in pre-pregnancy BMI corresponded to a lower weight gain during pregnancy, probably linked to a greater dietary and behavioral control [14]. However, BMI at delivery was significantly higher in diabetic patients; dietary behavioral control was apparently not sufficient to reverse the differences with the control group. No statistically significant differences were found for the threatened preterm labor, abnormal placental insertion and detachment, and amniotic fluid disorders. A higher frequency of threatened preterm labor was reported in the PGDM group, confirming the findings of Kong L. et al. [15]. This difference was more evident when preterm delivery was considered in patients with PGDM, explained by spontaneous onset, early induction of childbirth, prevention of maternal and/or fetal complications, and reduction of perinatal mortality. The most important causes of preterm childbirth in DM1 could be the uterine overdistension due to fetal macrosomia and/or polyhydramnios. Dollberg et al. [16] did not associate the high incidence of preterm childbirth with polyhydramnios, but recognized the role played by genitourinary infections and a history of previous preterm deliveries.

Regarding fetal outcomes, fetal macrosomia was found in the group of cases (12.1%).

Diabetic patients delivered more frequently by CS [17], including 90.5% of the DM1 patients.

No statistically significant differences were found between cases and controls in the use of medical induction of labor with prostaglandins or oxytocin. This is consistent with the fact that diabetic patients are often subjected to elective CS.

LGA infants were more frequently described in DM1 patients [18,19]. Another large population-based study in Catalonia [20] found a more prevalent LGA in infants of DM1 mothers. However, no relationship was found with the number of macrosomic fetuses, in line with the literature [21]: the error in the estimation of the fetal weight (10–15%) increases as the gestational age advances, and as the fetal weight increases [21].

Morphological abnormalities at birth have been documented for mothers with diabetes; in particular congenital, heart disease is the outcome most associated with diabetes mellitus [22,23].

The epidemiology of respiratory disorders at birth seems to partially differ from that of the scientific literature [24,25,26]. TTN rate was 3.4% in our cohort vs. 10% of other studies. The relationship between RDS and PGDM was confirmed in a recent meta-analysis by Yan Li et al. [27]. The incidence of neonatal hypoglycemia was higher in neonates of DM1 mothers (66.7%) [27,28,29]. The percentage of neonatal jaundice in children of diabetic mothers ranged from 8.7% to 29% [26,27,29]. Our study showed a higher incidence (74.1%), even if those requiring phototherapy were less frequent (44.8%).

Although the retrospective nature of the study and the selection associated to the enrollment in a reference center may hinder the statistical inference of the findings, the present study shows the epidemiological perspective of an Italian region characterized by a higher incidence of diabetes mellitus than in the general population. Some key limitations related to the observational and retrospective nature of the study should be acknowledged: a formal computation of the sample size was not performed for cases and controls, affecting the statistical power of some results; furthermore, the enrollment of the controls was consecutive, and no matching for the main confounding variables was carried out. Furthermore, some of the findings might be driven by social, epidemiological, and economic determinants; unfortunately, the medical files do not include such data that could be helpful to perform stratified analyses. For this reason, it is of paramount importance to early detect patients at risk to immediately implement preventive measures.

## Figures and Tables

**Table 1 jpm-12-01320-t001:** Anthropometric characteristics of the patients.

	*Controls (n = 116)*	*PGDM* *(n = 58)*	*DM1* *(n = 42)*	*DM2* *(n = 14)*	*p Value* *(Controls x PGDM)*	*p Value* *(Controls x DM1)*	*p Value* *(Controls x DM2)*
Median (IQR) pregestational weight, kg	57 (50.5–63)	61 (55.5–72.5)	60 (55–68)	76 (55.5–83.5)	0.003	ns	0.009
Median (IQR) pregestational BMI, kg/m^2^	22 (19.8–24)	23.7 (20.8–28)	23.7 (20.8–25.9)	28.2 (21.2–32.9)	0.005	0.01	0.03
Normal weight, *n* (%)	93 (80.2)	25 (52.1)	20 (58.8)	3 (25.0)	<0.0001	0.02	<0.0001
Underweight, *n* (%)	7 (6.0)	2 (4.2)	0 (0.0)	2 (16.7)	ns	ns	ns
Overweight, *n* (%)	14 (12.1)	10 (20.8)	8 (23.5)	2 (16.7)	ns	ns	ns
Obese, *n* (%)	3 (2.6)	8 (16.7)	3 (8.8)	5 (41.7)	0.001	ns	<0.0001
Median (IQR) weight increase, kg	12 (10–14)	11 (7.5–14)	11.5 (9–13)	11 (6–15.5)	ns	ns	ns
Excessive weight increase (>12 kg), *n* (%)	44 (37.9)	19 (39.6)	14 (41.2)	4 (33.3)	ns	ns	ns
Median (IQR) weight at delivery, kg	70 (63–75)	74 (65–85)	73 (65–80)	84.5 (65.0–92.5)	ns	ns	0.02
Median (IQR) BMI at delivery, kg/m^2^	26.6 (24.8–28.8)	27.9 (25.7–32.2)	27.8 (25.8–30.1)	30.6 (25.1–37.1)	ns	ns	ns

Pregestational diabetes mellitus (PGDM); diabetes mellitus type 1 (DM1); diabetes mellitus Type 2 (DM2); normal weight (BMI between 18 and 24.9 kg/m^2^); underweight (BMI < 18 kg/m^2^); overweight (BMI between 25 and 29.9 kg/m^2^); obese (BMI ≥ 30 kg/m^2^).

**Table 2 jpm-12-01320-t002:** Neonatal outcomes.

	*Controls (n = 116)*	*PGDM* *(n = 58)*	*DM1* *(n = 42)*	*DM2* *(n = 14)*	*p Value* *(Controls x PGDM)*	*p Value* *(Controls x DM1)*	*p Value* *(Controls x DM2)*
Fetus appropriate for gestational age (AGA), n (%)	96 (82.8)	27 (46.6)	17 (40.5)	8 (57.1)	<0.0001	<0.0001	0.04
Fetus small for gestational age (SGA), n (%)	12 (10.3)	4 (6.9)	1 (2.4)	3 (21.4)	ns	ns	ns
Fetus large for gestational age (LGA), n (%)	8 (6.9)	27 (46.5)	24 (57.1)	3 (21.4)	<0.0001	<0.0001	ns
Respiratory disorders, n (%)	11 (9.5)	18 (31.0)	15 (35.7)	3 (21.4)	<0.0001	<0.0001	ns
RDS (respiratory distress syndrome), n (%)	6 (5.2)	17 (29.3)	14 (33.3)	3 (21.4)	<0.0001	<0.0001	ns
TTN (neonatal transient tachypnea), n (%)	2 (1.7)	2 (3.5)	2 (4.8)	0 (0.0)	ns	ns	ns
Neonatal intubation, n (%)	4 (3.5)	6 (10.3)	6 (14.3)	0 (0.0)	ns	0.02	ns
Median (IQR) glycemia at 3 h, mg/dL	66.5 (60–73.5)	58.5 (37–72)	58 (37–76)	53 (37–67)	0.003	0.02	0.003
Median (IQR) lower glycemia, mg/dL	61.9 (14.2)	40.3 (17.0)	40.2 (18.1)	41.1 (13.4)	<0.0001	<0.0001	<0.0001
Neonatal hypoglycemia, n (%)	6 (5.2)	38 (65.5)	28 (66.7)	9 (64.3)	<0.0001	<0.0001	<0.0001
Neonatal jaundice, n (%)	29 (25.0)	43 (74.1)	43 (74.1)	9 (64.3)	<0.0001	<0.0001	0.002
Phototherapy, n (%)	14 (12.1)	26 (44.8)	26 (44.8)	4 (28.6)	<0.0001	<0.0001	ns
Morphological anomalies, n (%)	16 (13.8)	19 (32.8)	16 (38.1)	2 (14.3)	0.003	0.001	ns

**Table 3 jpm-12-01320-t003:** Number of cases with morphological anomalies.

*PGDM n = 19 (32.8%)*
Cardiovascular System
2 cases of patent foramen ovale with hemodynamically significant shunt
2 cases of atrial septal defect (ASD) type “ostium secundum”, associated with patency of the ductus arteriosus
2 cases of ventricular septal defect (VSD) with left-right shunt
1 case of mild mitral insufficiency
1 case of left ventricular hypertrophy
1 case of biventricular hypertrophy
Urogenital apparatus
2 cases of hydrocele
1 case of bilateral pyelectasis
1 case with left ovarian mass (21 mm ×18 mm)
1 case of left inguinal hernia
Other anomalies
2 cases of cleft palate
1 case of umbilical hernia associated with left inguinal hernia
1 case of thyroglossal duct cyst (in the same newborn with mild mitral insufficiency)
1 case of flat angioma on the forehead
** *Controls n = 16 (13.8%)* **
Cardiovascular system
2 cases of patent foramen ovale, with no hemodynamically significant shunt
1 case of sub-aortic ventricular septal defect (VSD)
1 case of mid-apical VSD with the presence of a mild shunt
1 case of patency of the ductus arteriosus with the presence of a left-right shunt
Musculoskeletal system
3 cases of congenital clubfoot (talipes equinovarus)
1 case of mandibular hypoplasia
1 case of clinodactyly in the 5th toe of the left foot
Central nervous system
1 case of hydrocephalus due to congenital stenosis of aqueduct of Sylvius (HSAS)
Urogenital apparatus
1 case of hypospadias
1 case of hydrocele
Genetic defects
2 cases of trisomy 21 (Down syndrome)
Other anomalies
1 cases of umbilical hernia

## Data Availability

The authors keep data that are available on request.

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
