# Peer review of "Fetal Growth and Neonatal Outcomes in Pregestational Diabetes Mellitus in a Population with a High Prevalence of Diabetes"

_jpm, 2022, doi:10.3390/jpm12081320_

Round 1

Reviewer 1 Report

Language revision is needed. As for neonatal outcome I don't understand why authors decided to use Apgar at the first minute and not the 5th, which appears to be more reliable. It is strange why there is no difference between the two groups as for induction matter. I would expect a higher rate in the PGDM group. I wonder which is the CS rate in the hospital and how many PGDM patients are induced and how many go to elective CS. Authors should clarify these data.

Author Response

Reviewer 1

Reviewer wrote (RW): Language revision is needed.

Answer of Authors (AA): The manuscript has been revised by a skilled native speaker.

RW: As for neonatal outcome I don't understand why authors decided to use Apgar at the first minute and not the 5th, which appears to be more reliable.

AA: We agree with the Reviewer, and we also reported 5th minute Apgar score. We recorded both.

RW: It is strange why there is no difference between the two groups as for induction matter:

AA: There are differences between the groups, but they are not statistically significant, probably because the sample is not large enough.

RW: I would expect a higher rate in the PGDM group. I wonder which is the CS rate in the hospital and how many PGDM patients are induced and how many go to elective CS.

AA: CS rate in primary cesarean surgery in our hospital is 34%. Induction of labor was performed in 10.9% of controls VS. 28.6% of PDGM group (p-value was not statistically significant, and its value was not included in the text). Prevalence of elective CS in the PDGM group was 37.3%.

Reviewer 2

RW: The main problem is that this study does not verify the effect of obesity on the results. The results observed in this population may have been influenced by maternal BMI and other factors such as glycemic control during pregnancy and metabolic control at conception.

AA: We thank the Reviewer for having raised this point. Table 1 includes key information.

RW: Please describe that this is a retrospective study, indicate the study location and how the data were collected:

AA: We included the requested information: “The aim of this retrospective study, conducted in an Italian tertiary care hospital”… “Data on pregnancy, childbirth, and newborns were collected from medical records.”

RW: This section is very short and could be improved. Explain the scientific basis and rationale for the investigation:

AA: Impaired glucose metabolism could be a risk factor for pregnancy complications and may be associated with adverse neonatal outcomes [1-4]; this study evaluated maternal-fetal and neonatal clinical outcomes of a cohort of patients with PGDM (DM1, DM2, and maturity onset diabetes of the young -MODY-) in comparison with those of pregnant individuals without diabetes.

The aim of the study was to describe pregnancy complications in patients with PDGM living in a Italian region with high prevalence.

RW: Describe the strategy used to recruit or include cases in the study. Clearly describe the inclusion and exclusion criteria. Could the authors inform whether twin pregnancies or cases of stillbirth were included?

AA: Inclusion criteria were pregnant women with PGDM (i.e., DM1 and DM2). The exclusion criteria were diagnosis of gestational diabetes, twin pregnancy, stillbirth.

RW: In the methods, enter the amount of glucose used in the OGTT test:

AA: “…and with a negative 75 g oral glucose test tolerance (75-g OGTT) performed at 24-28 weeks of gestation”

RW: Was the control group matched?

AA: “The two groups were matched by age (calculated at the time of delivery), with a ratio of 1:2 (PDGM:Controls).”

RW: Please do not start a sentence with a number

AA: “Among the patients in the PGDM group, 77.6% had a normal glycemia at the time of delivery…”

RW: Do not repeat the results presented in the tables in the Text

AA: A higher median (IQR) pre-gestational body weight [61 (55.5-72.5) VS. 57 (50.5-63); p-value: 0.003] and BMI [23.7 (20.8-28) VS. 22 (19.8-24); p-value: 0.005] were found in the PGDM group (Table 1). Furthermore, the prevalence of obesity was significantly higher among the cases. Similarly, RDS, as well as neonatal hypoglycemia, occurred more frequently in the PGDM group. Incidence of neonatal jaundice was significantly higher (74.1%) in the PGDM cohort. There was a statistically significant difference in birth weight centile values classified into fetus AGA (appropriate for gestational age), SGA (small for gestational age) and LGA (large for gestational age). The proportion of AGA was significantly higher in the control group (82.8% VS. 46.6%; p-value <0.0001). LGA prevalence was significantly higher in the PGDM group (46.5% VS. 6.9%; p-value <0.0001), especially between the control and DM1 group (6.9% VS. 57.1%; p-value <0.0001).

RW: Please describe the observed fetal malformations in a grouped manner, without listing the cases. Present the list of cases in the supplementary material:

AA: We created a new table (Table 3) and the individual information was included in the supplementary material as suggested.

RW: I suggest creating tables including columns: Controls /PGDM / DM1 / DM2 / p (controls x PGDM) / p (controls xDM1) / p (controls x DM2). I suggest changing the page layout.

AA: We edited tables 1 and 2

RW: Authors are requested to verify that all references have been presented in accordance with the journal's rules.

AA: We edited the list of references following the journal’s rules.

Reviewer 2 Report

This is a well-written article with adequate review of literature, well presentation and statistical analysis of the data.

Author Response

(The authors gave the same response as above.)

Reviewer 3 Report

Thank you for the opportunity to read this manuscript on fetal growth and neonatal outcomes in pregnancies complicated by pregestational diabetes mellitus.

The main problem is that this study does not verify the effect of obesity on the results. The results observed in this population may have been influenced by maternal BMI and other factors such as glycemic control during pregnancy and metabolic control at conception.

Abstract:

Please describe that this is a retrospective study, indicate the study location and how the data were collected.

Main text:

Introduction:

This section is very short and could be improved. Explain the scientific basis and rationale for the investigation.

Methods:

Describe the strategy used to recruit or include cases in the study. Clearly describe the inclusion and exclusion criteria.

Could the authors inform whether twin pregnancies or cases of stillbirth were included?

In the methods, enter the amount of glucose used in the OGTT test.

Was the control group matched?

Results:

Please do not start a sentence with a number.

Do not repeat the results presented in the tables in the text.

Please describe the observed fetal malformations in a grouped manner, without listing the cases. Present the list of cases in the supplementary material.

I suggest creating tables including columns: Controls / PGDM / DM1 / DM2 / p (controls x PGDM) / p (controls x DM1) / p (controls x DM2). I suggest changing the page layout.

Authors are requested to verify that all references have been presented in accordance with the journal's rules.

Author Response

Reviewer 2

RW: The main problem is that this study does not verify the effect of obesity on the results. The results observed in this population may have been influenced by maternal BMI and other factors such as glycemic control during pregnancy and metabolic control at conception.

AA: We thank the Reviewer for having raised this point. Table 1 includes key information.

RW: Please describe that this is a retrospective study, indicate the study location and how the data were collected:

AA: We included the requested information: “The aim of this retrospective study, conducted in an Italian tertiary care hospital”… “Data on pregnancy, childbirth, and newborns were collected from medical records.”

RW: This section is very short and could be improved. Explain the scientific basis and rationale for the investigation:

AA: Impaired glucose metabolism could be a risk factor for pregnancy complications and may be associated with adverse neonatal outcomes [1-4]; this study evaluated maternal-fetal and neonatal clinical outcomes of a cohort of patients with PGDM (DM1, DM2, and maturity onset diabetes of the young -MODY-) in comparison with those of pregnant individuals without diabetes.

The aim of the study was to describe pregnancy complications in patients with PDGM living in a Italian region with high prevalence.

RW: Describe the strategy used to recruit or include cases in the study. Clearly describe the inclusion and exclusion criteria. Could the authors inform whether twin pregnancies or cases of stillbirth were included?

AA: Inclusion criteria were pregnant women with PGDM (i.e., DM1 and DM2). The exclusion criteria were diagnosis of gestational diabetes, twin pregnancy, stillbirth.

RW: In the methods, enter the amount of glucose used in the OGTT test:

AA: “…and with a negative 75 g oral glucose test tolerance (75-g OGTT) performed at 24-28 weeks of gestation”

RW: Was the control group matched?

AA: “The two groups were matched by age (calculated at the time of delivery), with a ratio of 1:2 (PDGM:Controls).”

RW: Please do not start a sentence with a number

AA: “Among the patients in the PGDM group, 77.6% had a normal glycemia at the time of delivery…”

RW: Do not repeat the results presented in the tables in the Text

AA: A higher median (IQR) pre-gestational body weight [61 (55.5-72.5) VS. 57 (50.5-63); p-value: 0.003] and BMI [23.7 (20.8-28) VS. 22 (19.8-24); p-value: 0.005] were found in the PGDM group (Table 1). Furthermore, the prevalence of obesity was significantly higher among the cases. Similarly, RDS, as well as neonatal hypoglycemia, occurred more frequently in the PGDM group. Incidence of neonatal jaundice was significantly higher (74.1%) in the PGDM cohort. There was a statistically significant difference in birth weight centile values classified into fetus AGA (appropriate for gestational age), SGA (small for gestational age) and LGA (large for gestational age). The proportion of AGA was significantly higher in the control group (82.8% VS. 46.6%; p-value <0.0001). LGA prevalence was significantly higher in the PGDM group (46.5% VS. 6.9%; p-value <0.0001), especially between the control and DM1 group (6.9% VS. 57.1%; p-value <0.0001).

RW: Please describe the observed fetal malformations in a grouped manner, without listing the cases. Present the list of cases in the supplementary material:

AA: We created a new table (Table 3) and the individual information was included in the supplementary material as suggested.

RW: I suggest creating tables including columns: Controls /PGDM / DM1 / DM2 / p (controls x PGDM) / p (controls xDM1) / p (controls x DM2). I suggest changing the page layout.

AA: We edited tables 1 and 2

RW: Authors are requested to verify that all references have been presented in accordance with the journal's rules.

AA: We edited the list of references following the journal’s rules.

Round 2

Reviewer 3 Report

Thank you for the opportunity to read the revised version of this manuscript. I have minor questions:

Table 3 is not very clear to demonstrate the FETAL anomalies observed in the population studied. It is not clear whether the authors refer to the case number or the number of cases with the anomaly. It is suggested to create a more explanatory table or chart, or to indicate in the text in number and % of the total of each group. The title of the table should be improved.

Author Response

RESPONSE TO REVIEWER

Reviewer 3

Reviewer wrote (RW):

Thank you for the opportunity to read the revised version of this manuscript. I have minor questions:

Table 3 is not very clear to demonstrate the FETAL anomalies observed in the population studied. It is not clear whether the authors refer to the case number or the number of cases with the anomaly. It is suggested to create a more explanatory table or chart, or to indicate in the text in number and % of the total of each group. The title of the table should be improved.

Answer of Authors (AA):

We agree with the Reviewer. We referred to number of cases with the anomaly. We have now improved the title of the table: Table 3. Number of cases with morphological anomalies. PGDM n:19 (32.8%); Controls n:16 (13.8%)

This manuscript is a resubmission of an earlier submission. The following is a list of the peer review reports and author responses from that submission.

Round 1

Reviewer 1 Report

This was a retrospective study analyzing neonatal outcomes of newborns of mothers with pregestational diabetes in a specific population with a high prevalence of diabetes.

Line 46:           are instead of were

Line 125:         „77.6% of patients in the PGDM group had a good glycemic compensation at the time 125 of delivery.“ What exactly is meant by good glycemic compensation. I advise the authors to declare the criteria for that and when was the examination done.

Line 165:         „There was a statistically significant difference in birth weight.“ Then I think the authors should provide the readers with numbers.

Line 189:         „No statistically significant differences were found between cases and controls in the use of medical induction of labor with prostaglandins or oxytocin.“ These results are not even mentioned in the results chapter…

Comments:

  • The selection of control group in retrospective studies is essential. The authors should specify the mechanism by which they selected these subjets
  • In tables if the p-value is not significant, i tis possible to write NS instead of precise value, but I reccomend the authors to stick to one way of reporting thoughout the tables
  • 1. pregnancy disorders subchapter: Again it is not clear to me, how some outcomes such as „threatened miscarriage“ or „threatened preterm delivery“ were defined. Line 131 reports the incidence of „threatened preterm delivery“, while the line 133 reports „preterm delivery“… I advise the authors to add additional table summarizing perinatal outcomes.
  • The spelling should be corrected in line with the instructions for authors. For example, sometimes there is „foetal“, and sometimes „fetal“.
  • „Fetal echocardiography was used to investigate cardiac abnormalities more frequently in the PGDM group“. I pressume this means, that examination was requested, not that there was pathological finding.
  • In methods, the autors claim to collect information about Intrauterine growth retardation (IUGR) fetus, however, after that I have not found any informationn about it in the results. Also the right term should be Fetal growth restricion (FGR). I advise the authrors to ommit this and adhere soleley to SGA.
  • The number of morphologicla anomalies is rather higher in both control and PGDM groups. Perhaps these conditions deservessome closer description…
  • I feel that the title of the manuscript does not correspond to the content. With the exception of the prevalence of LGA, SGA, the paper does not deal in detail with the fetal growth. Moreover, the authors acknowledge that their results apply to a specific population with a high prevalence of diabetes. This should be reflected in the title.

I find serious flaws in the submitted article. First of all, not enough cases were included to accurately assess the incidence of complications. Furthermore, it is not clear how the control group was selected. The results are consistent with previously published papers and provide little new information on the topic. On the other hand, they may serve to counsel pregnant women from the aforementioned high-risk population with a high prevalence of diabetes about neonatal risks and may therefore be of clinical relevance.

Author Response

Manuscript ID: JCM (https://www.mdpi.com/journal/jcm) (ISSN 2077-0383) jcm-1637023

Article title: Fetal growth and neonatal outcomes in pregestational diabetes mellitus

AUTHORS (A): Giampiero Capobianco * , ALESSANDRA GULOTTA , GIULIO TUPPONI , Francesco Dessole , GIUSEPPE VIRDIS , Claudio Cherchi , Davide De Vita , Marco Petrillo , Giorgio Olzai , Roberto Antonucci , Laura Saderi , PIER LUIGI CHERCHI , Salvatore Dessole , Giovanni Sotgiu

Obstetrics & Gynecology (https://www.mdpi.com/journal/jcm/sections/Obstetrics_Gynecology)

Fetal Growth: What Is New in the Clinical Research? (https://www.mdpi.com/journal/jcm/special_issues/fetal_growth)

The aim of the study was to evaluate maternal-fetal and neonatal clinical outcomes in a group of patients with pregestational diabetes mellitus (PGDM) such as diabetes mellitus type 1 (DM1), diabetes mellitus type 2 (DM2), and maturity onset diabetes of the young (MODY). Overall, 174 pregnant women, nulliparous and multiparous, with single pregnancy were enrolled. The selected patients were divided into two groups: PGDM (42 with DM1, 14 with DM2, and 2 with MODY); 116 patients with a negative pathological history of diabetes mellitus were the control. We reported an incidence of preterm delivery of 55.2% in the PGDM group, of 59.5% in the DM1 group, and 42.9% in the DM2 group VS. 6% in the controls. Fetal growth disorders, such as intrauterine growth retardation, small for gestational age, and fetal macrosomia were found in 19% and 3.6% in the case and in the control group, respectively. A relationship between DM2 and gestational hypertension was found.

Author's Reply to the Review Report (Reviewer 1)

We provide a point-by-point response to the reviewer’s comments and either enter it in the box below

Reviewer 1 (R1)

This was a retrospective study analyzing neonatal outcomes of newborns of mothers with pregestational diabetes in a specific population with a high prevalence of diabetes.

Line 46: are instead of were 

Line 125: „77.6% of patients in the PGDM group had a good glycemic compensation at the time 125 of delivery.“ What exactly is meant by good glycemic compensation. I advise the authors to declare the criteria for that and when was the examination done. 

Authors (AA)

Line 46: we have now replaced “are” with “were”

Line 125: in the revised version (red ink) we have now written:

77.6% of patients in the PGDM group had a normal glycemia at the time of delivery according to the values established by the American Diabetes Association (fasting glucose lower than 90 mg/dl and Hb glycated lower than 6%) [11].

R1

Line 165: „There was a statistically significant difference in birth weight.“ Then I think the authors should provide the readers with numbers. 

Line 189: „No statistically significant differences were found between cases and controls in the use of medical induction of labor with prostaglandins or oxytocin.“ These results are not even mentioned in the results chapter...

AA

Line 165: now we have written: There was a statistically significant difference in birth weight centile values classified into fetus AGA (appropriate for gestational age), SGA (small for gestational age) and LGA (large for gestational age): The proportion of AGA was significantly higher in the control than in the PGDM group (82.8% VS. 46.6%; p-value <0.0001), regardless of diabetes type 1 or 2. LGA prevalence was significantly higher in the PGDM group than in the control group (46.5% VS. 6.9%; p-value <0.0001), especially between the control and DM1 group (6.9% VS. 57.1%; p-value <0.0001).

Line 189: now we have mentioned in subchapter 3.3 Pregnancy outcomes (results): PGDM group did not show a higher frequency of medical induction of vaginal delivery than that of the control group.

R1

The selection of control group in retrospective studies is essential. The authors should specify the mechanism by which they selected these subjects.

AA

We thank the Reviewer for this important methodological question. However, based on the observational and retrospective nature of the study we recruited consecutively only healthy individuals. We could not perform any matching for the main confounding variables and, following this question, we acknowledge this limitation in the Discussion section:

Some key limitations related to the observational and retrospective nature of the study should be acknowledged: a formal computation of the sample size was not performed for cases and controls, affecting the statistical power of some results; furthermore, the enrollment of the controls was consecutive and no matching for the main confounding variables was carried out. Furthermore, some of the findings might be driven by social, epidemiological, and economic determinants; unfortunately, the medical files do not include those data which could be helpful to perform stratified analyses.

R1

  1. a) In tables if the p-value is not significant, it is possible to write NS instead of precise value, but I recommend the authors to stick to one way of reporting thoughout the tables

AA

The requested changes were performed

R1

  1. pregnancy disorders subchapter: Again it is not clear to me, how some outcomes such as „threatened miscarriage“ or „threatened preterm delivery“ were defined. Line 131 reports the incidence of „threatened preterm delivery“, while the line 133 reports „preterm delivery“... I advise the authors to add additional table summarizing perinatal outcomes.

AA

We have reported in the text the definition of threatened miscarriage“ and threatened preterm delivery“

R1

The spelling should be corrected in line with the instructions for authors. For example, sometimes there is „foetal“, and sometimes „fetal“.

AA

We adopted fetal and not foetal

R1

Fetal echocardiography was used to investigate cardiac abnormalities more frequently in the PGDM group“. I presume this means, that examination was requested, not that there was pathological finding.

AA

We thank the Reviewer. To avoid any misunderstandings we edited the text: Fetal echocardiography was requested more frequently in the PGDM than in the control group (67.4% VS. 17%; p-value <0.0001) to detect fetal cardiac abnormalities.

R1

In methods, the authors claim to collect information about Intrauterine growth retardation (IUGR) fetus, however, after that I have not found any information about it in the results. Also the right term should be Fetal growth restriction (FGR). I advise the authors to ommit this and adhere soleley to SGA.

AA

We decided to follow the suggestion using FGR

R1

The number of morphologic anomalies is rather higher in both control and PGDM groups. Perhaps these conditions deserves some closer description...

AA

All the morphological anomalies are now reported in the text

R1

I feel that the title of the manuscript does not correspond to the content. With the exception of the prevalence of LGA, SGA, the paper does not deal in detail with the fetal growth. Moreover, the authors acknowledge that their results apply to a specific population with a high prevalence of diabetes. This should be reflected in the title.

AA

We edited the text as follows: Fetal growth and neonatal outcomes in pregestational diabetes mellitus in a population with a high prevalence of diabetes

R1

I find serious flaws in the submitted article. First of all, not enough cases were included to accurately assess the incidence of complications. Furthermore, it is not clear how the control group was selected. The results are consistent with previously published papers and provide little new information on the topic. On the other hand, they may serve to counsel pregnant women from the aforementioned high-risk population with a high prevalence of diabetes about neonatal risks and may therefore be of clinical relevance.

AA

We thank the Reviewer for this important point. We did not perform a formal estimation of the sample size, based on a suitable primary statistical hypothesis. However, as highlighted by the Reviewer, our findings confirmed previous results described by other observational studies. The issue of the quantitative and qualitative features of the case and control group, respectively, could represent a limitation which we acknowledged in the Discussion section.

Reviewer 2 Report

There is not enough emphasis on what is new in the report or unexpected in your findings compared to what is already known. It largely seems to confirm what is already known but unexpected findings could be drawn out more to increase interest. . 

A more sophisticated statistical analysis may have assessed whether other factors such as socio ecconomic status ( which could affect nutritonal status for instance) may have confounded the results.

The article would be better shortened to a letter as it does not contain enough new findings to justify a journal article. New or unexpected findings are not highlighted and discussed in sufficient depth to justify publication.   Furthermore,, there are some limitations including that the statistical analysis does not account for confounding factors which may explain some of the differences  in outcomes between the two groups such as social class/poverty. The quality/sophistication  of the analysis thus does not justify a full journal article.  At times the explanation is not clear enough. How the control group was selected is not described. The methods section includes a very long list of variables written as free text which is tiring to read and could be better presented. 

Author Response

Manuscript ID: JCM (https://www.mdpi.com/journal/jcm) (ISSN 2077-0383) jcm-1637023

Article title: Fetal growth and neonatal outcomes in pregestational diabetes mellitus

AUTHORS (A): Giampiero Capobianco * , ALESSANDRA GULOTTA , GIULIO TUPPONI , Francesco Dessole , GIUSEPPE VIRDIS , Claudio Cherchi , Davide De Vita , Marco Petrillo , Giorgio Olzai , Roberto Antonucci , Laura Saderi , PIER LUIGI CHERCHI , Salvatore Dessole , Giovanni Sotgiu

Obstetrics & Gynecology (https://www.mdpi.com/journal/jcm/sections/Obstetrics_Gynecology)

Fetal Growth: What Is New in the Clinical Research? (https://www.mdpi.com/journal/jcm/special_issues/fetal_growth)

The aim of the study was to evaluate maternal-fetal and neonatal clinical outcomes in a group of patients with pregestational diabetes mellitus (PGDM) such as diabetes mellitus type 1 (DM1), diabetes mellitus type 2 (DM2), and maturity onset diabetes of the young (MODY). Overall, 174 pregnant women, nulliparous and multiparous, with single pregnancy were enrolled. The selected patients were divided into two groups: PGDM (42 with DM1, 14 with DM2, and 2 with MODY); 116 patients with a negative pathological history of diabetes mellitus were the control. We reported an incidence of preterm delivery of 55.2% in the PGDM group, of 59.5% in the DM1 group, and 42.9% in the DM2 group VS. 6% in the controls. Fetal growth disorders, such as intrauterine growth retardation, small for gestational age, and fetal macrosomia were found in 19% and 3.6% in the case and in the control group, respectively. A relationship between DM2 and gestational hypertension was found.

Author's Reply to the Review Report (Reviewer 2)

We provide a point-by-point response to the reviewer’s comments and either enter it in the box below

Reviewer 2 (R2) Comments and Suggestions for Authors (AA)

R2

There is not enough emphasis on what is new in the report or unexpected in your findings compared to what is already known. It largely seems to confirm what is already known but unexpected findings could be drawn out more to increase interest. A more sophisticated statistical analysis may have assessed whether other factors such as socio-economic status (which could affect nutritional status for instance) may have confounded the results. The article would be better shortened to a letter as it does not contain enough new findings to justify a journal article. New or unexpected findings are not highlighted and discussed in sufficient depth to justify publication.  Furthermore, there are some limitations including that the statistical analysis does not account for confounding factors which may explain some of the differences in outcomes between the two groups such as social class/poverty. The quality/sophistication of the analysis thus does not justify a full journal article. At times the explanation is not clear enough. How the control group was selected is not described. The methods section includes a very long list of variables written as free text which is tiring to read and could be better presented.

AA

We thank the Reviewer for having raised this point. We agree on the necessity of including more variables, which could play an important role in the occurrence of the outcome. However, the observational nature of the study and the limited information included in the medical files hindered this opportunity.

Round 2

Reviewer 1 Report

Line 126 "Fetal macrosomia (fetal growth ≥95° percentile) was found in foetuses of diabetic mothers (p-value < 0.0001), the majority of whom in the DM1 group"              I do not understand the sentence. I do not know the prevalence of fetal macrosomia, I have not found that in the text, neither in the table. The discussion mentioned 12,1% though...

Line 127 Fetal growth restriction (fetal growth < 5° percentile). This is not the definition of fetal growth restriction! If the authors believe it is, please provide adequate citation.

The same applies to fetal macrosomia fetal growth ≥95° percentile). In my point of view, the term fetal macrosomia implies fetal growth beyond a specific weight, usually 4,000 g, regardless of the fetal gestational age. 

However in the Methods you state:

For this study we used the definition of the Royal College of Obstetricians and Gynaecologists (RCOG) [12] which informs UK clinical practice, based on sonographic estimated fetal weight (EFW) measurement < 10th percentile to describe a fetus that has not reached its target weight. Patients were divided in three groups for comparison; fetuses with EFW below the 10th percentile for gestational age (SGA), fetuses with EFW > 10th percentile for gestation (AGA) and fetuses  > 90th percentile for gestation (LGA) according to the Alexander growth standard [13]

In Table 2 you state:

Fetus small for gestational age (SGA) n (%) 3 (21.4)

In the results you state:

Fetal growth restriction (FGR) (fetal growth < 5° percentile) was found more frequently in in the DM2 group (21.4%; p-value: 0.02).

In the abstract you state:

Fetal growth disorders, such as intrauterine growth retardation, small for gestational age, and fetal macrosomia were found in 19% and 3.6% in the case and in the control group, respectively.

The provided information do not match. It is very confusing and I think the manuscript contains mistakes.

Line 126 – „Foetuses„ I already pointed this out in the first review…

In the abstract you state:

"A relationship between DM2 and gestational hypertension was found."

I have not found this relationship explained in the text. Actually, I didn't even find an incidence of gestational hypertension.

I still have serious reservations about the methodology. If the authors report the incidence of FGR or macrosomia but do not use an internationally accepted definition, then the results are misleading and can be misinterpreted.

Although the authors have corrected the manuscript, not all objections have been taken into account to my satisfaction.The objections I have concern fundamental methodological issues. 

Author Response

Manuscript ID: JCM (https://www.mdpi.com/journal/jcm) (ISSN 2077-0383) jcm-1637023

Article title: Fetal growth and neonatal outcomes in pregestational diabetes mellitus

AUTHORS (A): Giampiero Capobianco * , ALESSANDRA GULOTTA , GIULIO TUPPONI , Francesco Dessole , GIUSEPPE VIRDIS , Claudio Cherchi , Davide De Vita , Marco Petrillo , Giorgio Olzai , Roberto Antonucci , Laura Saderi , PIER LUIGI CHERCHI , Salvatore Dessole , Giovanni Sotgiu

Obstetrics & Gynecology (https://www.mdpi.com/journal/jcm/sections/Obstetrics_Gynecology)

Fetal Growth: What Is New in the Clinical Research? (https://www.mdpi.com/journal/jcm/special_issues/fetal_growth)

The aim of the study was to evaluate maternal-fetal and neonatal clinical outcomes in a group of patients with pregestational diabetes mellitus (PGDM) such as diabetes mellitus type 1 (DM1), diabetes mellitus type 2 (DM2), and maturity onset diabetes of the young (MODY). Overall, 174 pregnant women, nulliparous and multiparous, with single pregnancy were enrolled. The selected patients were divided into two groups: PGDM (42 with DM1, 14 with DM2, and 2 with MODY); 116 patients with a negative pathological history of diabetes mellitus were the control. We reported an incidence of preterm delivery of 55.2% in the PGDM group, of 59.5% in the DM1 group, and 42.9% in the DM2 group VS. 6% in the controls. Fetal growth disorders, such as intrauterine growth retardation, small for gestational age, and fetal macrosomia were found in 19% and 3.6% in the case and in the control group, respectively. A relationship between DM2 and gestational hypertension was found.

Author's Reply to the Review Report (Reviewer 2)

We provide a point-by-point response to the reviewer’s comments and either enter it in the box below

Reviewer 1 (R1) Comments and Suggestions for Authors (AA)

R1

Line 126 “Fetal macrosomia (fetal growth > 95° percentile) was found in foetuses of diabetic mothers (p-value < 0.0001), the majority of whom in the DM1 group” do not understand the sentence. I do not know the prevalence of fetal macrosomia, I have not found that in the text, neither in the table. The discussion mentioned 12,1% though…

AA

We thank the Reviewer for having raised this point. We agree with you and now we have better specified in abstract (line 31):

Neonatal macrosomia (birth weight > 4000 g) was found in PGDM group (7 fetuses, 12.1%) and in no patients in the control group (p-value <0.0001); the majority of whom in the DM1 group (6 fetuses, 14.3%; p-value <0.0001).

R1

Line 127 Fetal growth restriction (fetal growth < 5° percentile). This is not the definition of Fetal growth restriction! If the authors believe it is, please provide adequate citation

AA

We are sorry for the mistake of printing (5 and not 10° percentile) but now we have inserted the right definition of FGR

We have now inserted the definition (line 77) related to this reference: Society for Maternal-Fetal Medicine (SMFM). Electronic address: [email protected], Martins JG, Biggio JR, Abuhamad A. Society for Maternal-Fetal Medicine Consult Series #52: Diagnosis and management of fetal growth restriction: (Replaces Clinical Guideline Number 3, April 2012). Am J Obstet Gynecol. 2020 Oct;223(4):B2-B17.

This sentence has been inserted in materials and methods: Fetal Growth Restriction (FGR) is defined as an ultrasonographic estimated fetal weight (EFW) or AC below the 10th percentile for gestational age (GRADE 1B)

R1

The same applies to fetal macrosomia fetal growth > 95° percentile). In my point of view, the term fetal macrosomia implies fetal growth beyond a specific weight, usually 4,000 g, regardless of the fetal gestational age. However in the Methods you state: For this study we used the definition of the Royal College of Obstetricians and Gyanecologists (RCOG) (12) which informs UK clinical practice, based on sonographic estimated fetal weight (EFW) measurement < 10th percentile to describe a fetus that has not reached its target weight. Patients were divided in three groups for comparison; fetuses with EFW below 10th percentile for gestational age (SGA), fetuses with EFW > 10th percentile for gestation (AGA) and fetuses > 90th percentile for gestation (LGA) according to Alexander growth standard (13)

AA

We agree with you and I report the definition of macrosomia that I have now added to my references:

Generally, fetal macrosomia may be defined by a birth weight >4000 g or higher cutoffs. Since a clear-cut definition of fetal macrosomia has not yet been established, a clinical value independent of gestational age, such as large for gestational age (LGA), is preferable. LGA fetuses are usually defined as those with a birth weight >90th percentile for gestational age. Reference: Araujo Júnior E, Peixoto AB, Zamarian AC, Elito Júnior J, Tonni G. Macrosomia. Best Pract Res Clin Obstet Gynaecol. 2017 Jan;38:83-96. 

R1

In table 2 you state:

Fetus small for gestational age (SGA) n (%) 3 (21.4%)

In the results you state:

Fetal growth restriction (FGR) (fetal growth < 5° percentile) was found more frequently in the DM2 group (21.4%; p-value: 0.02).

In the abstract you state:

Fetal growth disorders, such as intrauterine growth retardation, small for gestational age, and fetal macrosomia were found in 19% and 3.6% in the case and control group, respectively.

The provided information do not match. It is very confusing and I think the manuscript contains mistakes.

Lines 126 – “Foetuses”, I already pointed this out in the first review..

In the abstract you state:

“A relationship between DM2 and gestational hypertension was found”.

I have not found this relationship explained in the text. Actually, I didn’t even find an incidence of gestational hypertension.

I still have serious reservations about methodology. If the authors report the incidence of FGR or macrosomia but do not use an internationally accepted definition, then the results are misleading and can be misinterpreted.

Although the authors have corrected the manuscript, not all objections have been taken into account to my satisfaction. The objections I have concern fundamental methodological issues.

AA

We agree with you and are sorry to be confusing but now we have corrected in the manuscript:

Foetuses changed in the text with Fetuses

Abstract:

Fetal growth disorders, such as FGR, were more prevalent in the PGDM patients (11 fetuses, 19% PGDM VS 3 fetuses, 2.6% control group; p-value <0.0001). Neonatal macrosomia (birth weight > 4000 g) was found in PGDM group (7 fetuses, 12.1%) and in no patients in the control group (p-value <0.0001); the majority of whom in the DM1 group (6 fetuses, 14.3%; p-value <0.0001). Pregnancy-induced hypertension and preeclampsia were found only in the PGDM group: in 8 and 9 cases respectively (13.8% and 15.5%) and not in the control group (p <0.0001).
